# On the Anisotropy of Score-Based Generative Models

Andreas Floros [1]   Seyed-Mohsen Moosavi-Dezfooli [2]   Pier Luigi Dragotti [1]

## Abstract

We investigate the role of network architecture in shaping the inductive biases of modern score-based generative models. To this end, we introduce the *Score Anisotropy Directions* (SADs), architecture-dependent directions that reveal how different networks preferentially capture data structure. Our analysis suggests that SADs form adaptive bases aligned with the architecture's output geometry, providing a principled way to predict generalization ability in score models prior to training. Through both synthetic data and standard image benchmarks, we demonstrate that SADs reliably capture fine-grained model behavior and correlate with downstream performance, as measured by Wasserstein metrics. Our work offers a new lens for explaining and predicting directional biases of generative models.

## 1. Introduction

Neural networks generalize through inductive biases, i.e., biases that guide learning beyond training data (Wilson & Izmailov, 2020; Goyal & Bengio, 2022). For discriminative tasks, they are partially characterized through the *Neural Anisotropy Directions* (NADs), which reveal the architecture's directional preferences in the input space (Ortiz-Jimenez et al., 2020). However, generative modeling lacks a cohesive theory that explains how architectural geometry interacts with data manifolds (Kadkhodaie et al., 2024; An et al., 2025). In this work, we present a unified approach to explaining and interpreting inductive biases of score-based generative models by examining anisotropy in the output space, where networks exhibit preferential learning along certain directions. As motivated in the experiment of Figure 1, we posit that generalization ability is largely characterized by the alignment of the data with the architecture's geometry at initialization, which is denoted as $\mathbf{G}_{\mathcal{F}}$ here.

[1]Imperial College London [2]Apple. Correspondence to: Andreas Floros <andreas.floros18@imperial.ac.uk>.

*Proceedings of the 43rd International Conference on Machine Learning*, Seoul, South Korea. PMLR 306, 2026. Copyright 2026 by the author(s).

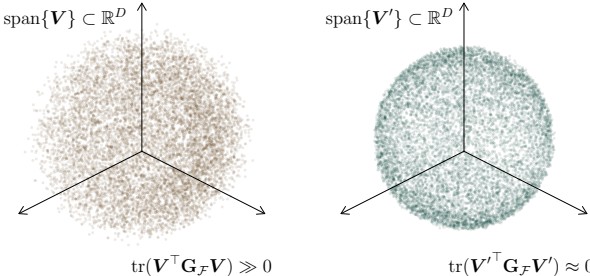

*Figure 1.* Sphere modeling in subspaces of $\mathbb{R}^D$ ($D = 256$) via DiT (Peebles & Xie, 2023). The only difference is the choice of subspace: on the left, data lives in a subspace aligned with the network's geometry, $\mathbf{G}_{\mathcal{F}}$, whereas the simulation on the right is in a non-aligned subspace. Despite identical training and sampling setups, quality differs consistently across repeated trials, suggesting that alignment with architectural geometry controls generalization.

Our contribution is making this notion of geometry precise and decomposing the output space in terms of anisotropy directions induced by it, i.e., the *Score Anisotropy Directions*.

> **Definition 1** (**Score Anisotropy Directions**).
> The *Score Anisotropy Directions* (SADs) of an architecture are the ordered set of orthonormal vectors of the output space, $\{\boldsymbol{u}_i\}_{i=1}^{D}$, ranked in terms of the preference of the network to generate data along those particular directions via score-based generative modeling in $\mathbb{R}^D$.

## 2. Background

We first give an overview of score-based generative models and provide necessary context on established methodology that examines inductive biases of deep neural networks.

### 2.1. Score-Based Generative Models

At the heart of score-based generative modeling is the *score function*, i.e., the gradient of the log-density of a data distribution $p$. With an arbitrary prior, $\boldsymbol{x}_0 \sim \pi$, and an estimate of the score, one samples from $p$ via Langevin dynamics:

$$\boldsymbol{x}_k = \boldsymbol{x}_{k-1} + \frac{\eta}{2} \overbrace{\nabla_{\boldsymbol{x}} \log p(\boldsymbol{x}_{k-1})}^{\text{"score"}} + \sqrt{\eta}\boldsymbol{z}_k, \qquad (1)$$

where $\boldsymbol{z}_k \sim \mathcal{N}(\boldsymbol{0}, \boldsymbol{I})$ is standard Gaussian and $\eta > 0$.

If $\eta \to 0$ and $K \to \infty$, under certain technical conditions, the iterates, $x_K$, in Equation 1 converge to a sample from $p$ (Welling & Teh, 2011). However, a practical limitation of the above setup is that estimated scores are inaccurate in low-density regions (e.g., in early iterations where learning is intractable). Moreover, score functions may be undefined in the case of data residing on low-dimensional manifolds. That is, the overall approach breaks down under the commonly adopted manifold hypothesis (Song & Ermon, 2019).

The research community has therefore largely moved on to Denoising Score Matching (DSM) (Vincent, 2011) and annealed Langevin dynamics, i.e., diffusion models, which are also our main focus.[1] More specifically, consider noise scales, $\sigma \in [\sigma_{\min}, \sigma_{\max}]$, and associated probability densities, $q_\sigma = p * \mathcal{N}(\mathbf{0}, \sigma^2 \mathbf{I})$, where $*$ represents convolution. Here, $\sigma_{\min}$ is small enough such that $q_{\sigma_{\min}} \approx p$ and $\sigma_{\max}$ is large enough so we can write $q_{\sigma_{\max}} \approx \mathcal{N}(\mathbf{0}, \sigma_{\max}^2 \mathbf{I})$. With estimates of scores of the perturbed distributions, $\nabla_x \log q_\sigma$, one samples from $p$ by decaying $\sigma$ from $\sigma_{\max}$ to $\sigma_{\min}$ (Ho et al., 2020; Song et al., 2021). This way, accurate modeling along sampling trajectories is tractable and the noise ensures support over the entirety of the ambient space, overcoming the limitations of naive Langevin dynamics. In particular, the scores are equivalent to minimum mean squared error Gaussian denoisers (Efron, 2011). That is, for neural networks, $\mathcal{F}_{\boldsymbol{\theta}} : \mathbb{R}^D \times \mathbb{R} \to \mathbb{R}^D$, parameterized by $\boldsymbol{\theta}$, one can approximate the scores via the following DSM optimization:

$$\min_{\boldsymbol{\theta}} \mathbb{E}_{x \sim p, \epsilon \sim \mathcal{N}(\mathbf{0}, \mathbf{I}), \sigma}[\hat{\mathcal{J}}_{\text{DSM}}(x, \epsilon, \sigma; \mathcal{F}_{\boldsymbol{\theta}})], \quad (2)$$

with $\hat{\mathcal{J}}_{\text{DSM}}(x, \epsilon, \sigma; \mathcal{F}_{\boldsymbol{\theta}}) := \left\| \mathcal{F}_{\boldsymbol{\theta}}(x + \sigma\epsilon, \sigma) + \frac{\epsilon}{\sigma} \right\|_2^2$.

## 2.2. Inductive Biases in Deep Learning

There is a vast literature on understanding the inductive biases of deep neural networks (Wilson & Izmailov, 2020; Goyal & Bengio, 2022). Of particular interest is the work of Ortiz-Jimenez et al. (2020), who identify directional biases in classifiers, i.e., the *Neural Anisotropy Directions* (NADs). More recently, Movahedi et al. (2025) extended the NAD framework by formalizing input-space architectural geometry and exploring its evolution during training. Specifically, they propose the *Geometric Invariance Hypothesis*, which, roughly, posits that NADs persist through training.

To our knowledge, contrary to the discriminative case, there is no unified theory on preferred modeling directions in the generative setting. Kadkhodaie et al. (2024) have argued that convolutional diffusion models are biased towards *Geometry-Adaptive Harmonic Bases* (GAHBs). However, they acknowledge that a mathematically precise definition of such bases remains an open question.

---

[1]Without loss of generality, we will standardize notation to the variance exploding formulation of Song & Ermon (2019).

More fundamentally, follow-up work by An et al. (2025) suggests that the GAHB framework may not extend to transformer-based diffusion models. In particular, they instead resort to a transformer-specific analysis by inspecting and manipulating attention maps.

In contrast to the above-mentioned works on the inductive biases of diffusion models, our goal is to present a more unified treatment of directional biases that is agnostic to the neural network's architecture. Our key insight is recognizing that the NAD framework can be adapted and extended to score-based modeling by assuming an underlying data log-density that assigns high probability to "on-manifold" data and low probability otherwise. With this approach, our analysis amounts to understanding anisotropy directions of such implicitly induced discriminative models.

## 3. Directional Biases of Diffusion Models

Central to our exploration of directional inductive biases in diffusion models is the following research question:

*Among equidimensional manifolds, which are preferred by diffusion modeling and how are preferences quantified?*

Toward answering the above, we consider data manifolds aligned with a particular direction, $v \in \mathbb{S}^{D-1}$, and investigate generalization ability as a function of the direction. Concretely, we will first study anisotropic Gaussian distributions of the form $\mathcal{N}(\mathbf{0}, vv^\top)$ with the objective of finding a suitable basis for $\mathbb{R}^D$ from which we can draw $v$ that reveal directional preferences. For a given basis, we independently train diffusion models on datasets formed by each normalized element under identical settings. Performance is quantified via the (Max-)Sliced Wasserstein $p$-distance, (M)SW$_p$, between the distribution obtained by sampling from the trained model, $\mu$, and the ground truth data, corresponding to $\nu = \mathcal{N}(\mathbf{0}, vv^\top)$. Notably, these are valid statistical distances and slice metrics are easily estimated via order statistics and Monte Carlo methods. Taking $p = 2$, the proposed metrics are mathematically defined as follows:

$$\text{SW}_2^2(\mu, \nu) = \mathbb{E}_{\boldsymbol{\theta} \sim \mathcal{U}(\mathbb{S}^{D-1})} \text{W}_2^2(\boldsymbol{\theta}_\#^\top \mu, \boldsymbol{\theta}_\#^\top \nu),$$
$$\text{MSW}_2^2(\mu, \nu) = \sup_{\boldsymbol{\theta} \in \mathbb{S}^{D-1}} \text{W}_2^2(\boldsymbol{\theta}_\#^\top \mu, \boldsymbol{\theta}_\#^\top \nu), \quad (3)$$
$$\text{W}_2^2(\boldsymbol{\theta}_\#^\top \mu, \boldsymbol{\theta}_\#^\top \nu) = \int_0^1 |F_{\boldsymbol{\theta}_\#^\top \mu}^{-1}(q) - F_{\boldsymbol{\theta}_\#^\top \nu}^{-1}(q)|^2 \mathrm{d}q,$$

where $\mathcal{U}$ denotes a uniform distribution, $F_{(\cdot)}^{-1}$ are quantile functions and $(\cdot)_\#$ represents the push-forward. Further details regarding our setup are included in Appendix A.

In Figure 2 we report our findings (averages over five runs) with the above-described approach over five common bases typically studied in signal processing literature.

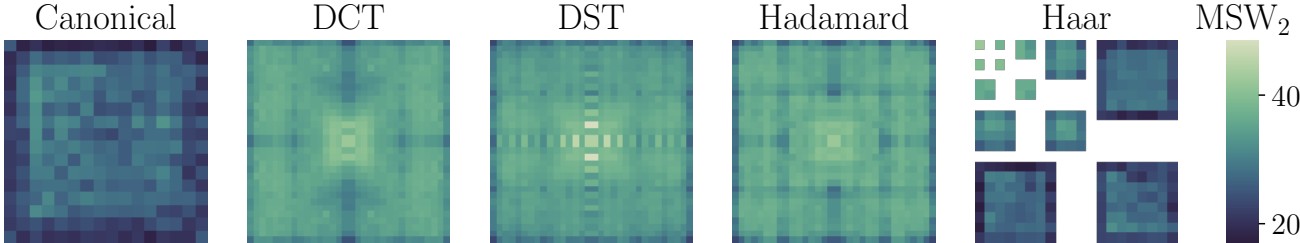

*Figure 2.* MSW$_2$ distance (computed over 10k test samples and 16384 projections) of iDDPM U-Net (Nichol & Dhariwal, 2021) architecture. Each pixel corresponds to a rank-one dataset of 16×16 images (with 10k training samples) aligned with a basis element of the canonical basis, DCT, DST, (ordered) Hadamard transform or Haar wavelet transform. That is, for a given location (canonical) or frequency / sequency (DCT, DST, Hadamard) or scale, channel and location (Haar), we visualize performance on the corresponding dataset. For ease of visualization, in the case of DCT, DST and Hadamard, we center the zero frequency dataset and extend the images to the left and top regions while respecting the symmetries of the transforms. See Appendix A for further implementation details.

We note two seemingly disconnected phenomena. Contrary to the conventional wisdom that neural networks better adapt at lower frequencies (Rahaman et al., 2019), the widely-adopted iDDPM U-Net architecture (Nichol & Dhariwal, 2021) struggles with low-frequency data. This is evidenced by the center points of the DCT, DST, Hadamard images and the top-left corner of the Haar image. Also, comparing results of the canonical and Haar bases with the frequency / sequency transforms, we see that vectors localized in space are better modeled, especially around the image borders.

Although the experiments of Figure 2 are insightful, it is unclear whether standard bases can reliably describe the biases of diffusion models given that they are completely decoupled from the underlying architecture. We argue that, in general, we cannot expect to uncover the intricacies of directional biases in this manner. Moreover, each considered basis amounts to training hundreds of diffusion models, which becomes impractical even for relatively small-dimensional data. We are therefore motivated to investigate a more principled and unified framework for directional biases.

### 3.1. Why Would Certain Directions Be Preferred?

While the learning process has a number of hyperparameters that potentially induce asymmetry, we focus on analyzing the role of the architecture. Prior work on biases of discriminative models argues preferences may emerge due to anisotropic loss of information or, in general, conditioning of the optimization landscape (Ortiz-Jimenez et al., 2020).

Indeed, such conditioning naturally manifests in the iDDPM U-Net architecture, e.g., via asymmetric resampling layers. We verify this in Figure 3, where we see that the default `nearest` interpolation leads to a clear directional bias. Similarly, one hypothesizes that border effects observed in canonical and Haar experiments of Figure 2 are attributed to the padding strategy employed in convolutional layers. Intuitively, one also expects that data living on the borders of images is poorly approximated by the network.

To more rigorously understand the effect of anisotropic conditioning, we now revisit the problem of learning rank-one distributions in a tractable setting, amenable to analysis. Specifically, for our Gaussian data, we note that linear DSM is sufficiently expressive, as demonstrated in Lemma 1. In this setup, we choose to model anisotropy explicitly, via a fixed transformation at the output with decaying eigenvalues. Under a (Stochastic) Gradient Descent, (S)GD, procedure, learning dynamics are then characterized by the following.

**Theorem 1** (Linear DSM dynamics, proof in Appendix C.4).
*Consider DSM with data drawn from $\mathcal{N}(\mathbf{0}, \mathbf{v}\mathbf{v}^\top)$ for noise level $\sigma > 0$ and $\mathbf{v} \in \mathbb{S}^{D-1}$. Let $\mathcal{F} : \mathbb{R}^D \to \mathbb{R}^D$ be linear networks expressed as $\mathbf{\Omega}(\cdot)$, where $\mathbf{\Omega} = \mathbf{\Phi}\mathbf{\Theta}$ with $\mathbf{\Phi}$ fixed. Denote the sorted eigenvalues of $\mathbf{\Phi}\mathbf{\Phi}^\top$ as $\{\lambda_i\}_{i=1}^D$ with $\lambda_{D-1} > \lambda_D > 0$ and corresponding normalized eigenvectors as $\{\mathbf{u}_i\}_{i=1}^D$. Assume a (S)GD procedure on initially zero-mean $\mathbf{\Theta}$, where the score is approximated by $\mathcal{F}$.*

*Choosing $\mathbf{v} = \mathbf{u}_i$, after $t$ steps, and with a sufficiently small learning rate $\eta > 0$, the mean error, $\mathbb{E}[\mathbf{\Omega}_t - \mathbf{\Omega}_*]$, to the optimal solution, $\mathbf{\Omega}_*$, converges as $\mathcal{O}[(1 - 2\eta\rho_i)^t]$ with $\rho_1 = \rho_i < \rho_D \; \forall i < D$. Moreover, near optimality, the SGD steps with respect to $\mathbf{\Theta}$ for $\mathbf{v} = \mathbf{u}_i$ have covariance $\mathrm{Cov}[\nabla_{\mathbf{\Theta}}\hat{\mathcal{J}}_{\mathrm{DSM}}] \propto \lambda_i$. That is, small eigenvalues imply stronger generalization compared to large eigenvalues.*

**Remark 1.** *If in the setup of Theorem 1 we let $\lambda_{D-1} = \lambda_D$, GD dynamics become isotropic. That is, the convergence rate of the mean is $\mathcal{O}[(1 - 2\eta\sigma^2\lambda_D)^t]$, independent of $i$.*

Curiously, the effect of anisotropic conditioning is limited in deterministic GD. Instead, our derivations suggest that stochasticity is the key ingredient that enables it. To confirm this, we design a small-scale experiment in $\mathbb{R}^5$, with the results shown in Figure 4. Specifically, for small $t$, the optimization dynamics are well-predicted by the deterministic GD analysis of Theorem 1 since, intuitively, the iterates are far from the optimum and the deterministic drift, i.e., the gradient, dominates the stochastic fluctuation. Consequently, all vectors except $\mathbf{u}_5$ yield similar results, as expected.

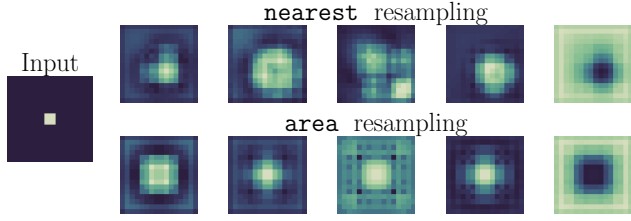

Input

nearest resampling

area resampling

*Figure 3.* Responses of the iDDPM architecture (Nichol & Dhariwal, 2021) with a symmetric initialization scheme. We show the default implementation, which uses `nearest` resampling layers, and a modified architecture that uses `area` resampling. We probe the models with a centered impulse input, shown on the left. Observe that the default resampling layers introduce asymmetry.

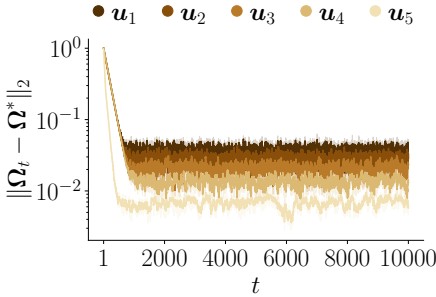

*Figure 4.* Setting of Theorem 1 ($\sigma = 1$) in $\mathbb{R}^5$ with SGD. For the eigenvectors, $\{u_i\}_{i=1}^5$, of $\boldsymbol{\Phi}\boldsymbol{\Phi}^\top$, we show errors between $\boldsymbol{\Omega}_t = \boldsymbol{\Phi}\boldsymbol{\Theta}_t$ and the optimal operator, $\boldsymbol{\Omega}^*$, defined in Lemma 1.

For large $t$, however, where the gradient norm is sufficiently small, noise dominates and it appears we have convergence to a stationary distribution. In such stochastic regimes, the error scales with the magnitude of the stochastic gradient covariance (Mandt et al., 2017), which we show to be $\propto \lambda_i$.

The takeaway from this discussion is that the best performance is achieved when the data is *not* aligned with the "geometry" that is induced by the score network, but instead lives in the subspace defined by its *smallest* eigenvalues.

### 3.2. Identifying the Score Anisotropy Directions

Having developed some intuition in tractable settings, we now extend and formalize these ideas more broadly, for potentially non-linear architectures and arbitrary distributions.

Concretely, for each noise level, $\sigma$, and assuming a (conservative and normalizable) parameterization via a neural network family, $\mathcal{F}$, we intuitively treat a realization, $\mathcal{F}_{\boldsymbol{\theta}} : \mathbb{R}^D \times \mathbb{R} \to \mathbb{R}^D$, as implicitly defining a log-density function, $\boldsymbol{x} \mapsto \log q_{\boldsymbol{\theta},\sigma}(\boldsymbol{x})$, that assigns high probability to $\boldsymbol{x}$ on the noise-perturbed data manifold and low probability to off-manifold data, i.e., we can write $\mathcal{F}_{\boldsymbol{\theta}}(\boldsymbol{x}, \sigma) \approx \nabla_{\boldsymbol{x}} \log q_{\boldsymbol{\theta},\sigma}(\boldsymbol{x})$. Here, we use the term "manifold" loosely, meaning regions of $\mathbb{R}^D$ that are statistically likely to be sampled by score-based modeling via $\mathcal{F}_{\boldsymbol{\theta}}$, i.e., the spanning set of the top SADs that we aim to uncover.

Intuitively, if one were to fix $\boldsymbol{x}$ and consider a small perturbation along some direction, $\boldsymbol{v}$, an abrupt change in the log-density indicates falling off or entering the manifold, that is, $\boldsymbol{v} = \boldsymbol{v}_\perp$ is perpendicular to the preferred modeling directions. Similarly, if $\boldsymbol{v} = \boldsymbol{v}_\parallel$ is parallel to the manifold, then we expect minimal changes in the log-density, i.e., we are traveling along a contour line. We refer the reader to Figure 6 for an illustration of our argument. Now, by taking the limit as the perturbation magnitude, $\tau$, tends to zero, we summarize these dynamics via directional derivatives, i.e., we can express changes in log-density via $|\log q_{\boldsymbol{\theta},\sigma}(\boldsymbol{x} + \tau\boldsymbol{v}) - \log q_{\boldsymbol{\theta},\sigma}(\boldsymbol{x})| \propto |\boldsymbol{v}^\top \nabla_{\boldsymbol{x}} \log q_{\boldsymbol{\theta},\sigma}(\boldsymbol{x})|$.

With this simplification, a straightforward application of Markov's inequality bounds the *a priori* probability of crossing the manifold by moving along $\boldsymbol{v}$, suggesting that directions attaining the minimum upper bound are inherently easier to model. That is, such directions are aligned with the prior densities induced by $\mathcal{F}_{\boldsymbol{\theta}}$. Specifically, for the family of networks, $\mathcal{F}$, parameterized by $\boldsymbol{\theta} \sim \Theta$ over noise levels, $\sigma$, and probing with $(\boldsymbol{x}, \sigma) \sim \mathcal{P}$, we write the bound:

$$\mathbb{P}\big(\big|\boldsymbol{v}^\top \nabla_{\boldsymbol{x}} \log q_{\boldsymbol{\theta},\sigma}(\boldsymbol{x})\big| \geq \eta\big) \leq$$
$$\boldsymbol{v}^\top \underbrace{\Big[\mathbb{E}_{(\boldsymbol{x},\sigma)\sim\mathcal{P},\,\boldsymbol{\theta}\sim\Theta}\mathcal{F}_{\boldsymbol{\theta}}(\boldsymbol{x},\sigma)\,\mathcal{F}_{\boldsymbol{\theta}}(\boldsymbol{x},\sigma)^\top\Big]}_{\text{"geometry"}} \boldsymbol{v}/\eta^2. \quad (4)$$

We view the above heuristic as a generalization of the conclusions of Section 3.1, recovering our previous findings as a special case. In particular, applying this bound on the setting of Theorem 1 with $\boldsymbol{\theta}$ iid and for any $\mathcal{P}$, the quantity over the underbrace, namely the *geometry*, recovers $\boldsymbol{\Phi}\boldsymbol{\Phi}^\top$, whose eigendecomposition defined the SADs in the case of linear networks. Moreover, the Markov bound correctly predicts that the small-eigenvalue vectors are easier to model compared to larger-eigenvalue eigenvectors. We therefore hope that this quantity also captures directional biases of more general architectures. We formalize this notion below.

**Definition 2** (**Average Geometry**).
Let $\mathcal{F}_{\boldsymbol{\theta}} : \mathbb{R}^D \times \mathbb{R} \to \mathbb{R}^D$ be a family of networks parameterized by $\boldsymbol{\theta}$. In the context of diffusion modeling, we define the average geometry of $\mathcal{F}$, induced by a probing distribution, $\mathcal{P}$, and a parameter distribution $\Theta$, as:

$$\mathbf{G}_{\mathcal{F}}(\mathcal{P}, \Theta) = \mathbb{E}_{\mathcal{P},\Theta}\big[\mathcal{F}_{\boldsymbol{\theta}}(\boldsymbol{x}, \sigma)\mathcal{F}_{\boldsymbol{\theta}}(\boldsymbol{x}, \sigma)^\top\big], \quad (5)$$

where $(\boldsymbol{x}, \sigma) \sim \mathcal{P}$. We assume $\mathcal{F}_{\boldsymbol{\theta}}$ is, roughly, aligned with some density, $q_{\boldsymbol{\theta},\sigma}$, via $\mathcal{F}_{\boldsymbol{\theta}}(\boldsymbol{x}, \sigma) \approx \nabla_{\boldsymbol{x}} \log q_{\boldsymbol{\theta},\sigma}(\boldsymbol{x})$.

Note, unlike the linear case investigated in Theorem 1, the notion of average geometry is, in general, local and adapts to the probe $\mathcal{P}$. However, in practice, we find that this is a non-critical hyperparameter and not central to our analysis.

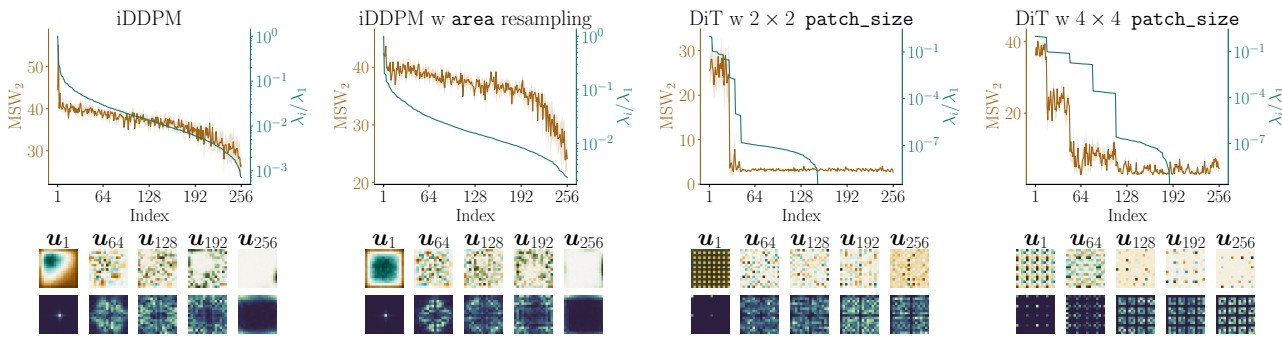

*Figure 5.* Test MSW$_2$ distance for different architectures on datasets aligned with the eigenvectors of their geometry at initialization, probing with $\mathcal{P} = \delta_{\mathbf{0}} \times \mathcal{U}(\{\sigma_{\min}, \ldots, \sigma_{\max}\})$, where $\delta$ denotes the Dirac delta. We report the mean $\pm$ the standard error over five independent runs. Corresponding normalized eigenvalues are on the right axes. The eigenvectors, with their energy in the DFT (zero frequency is centered), are shown below the plots (first, last row respectively). The experimental setup is identical to the one described in the beginning of Section 3 and Figure 2. We refer the reader to Appendix A for further implementation details.

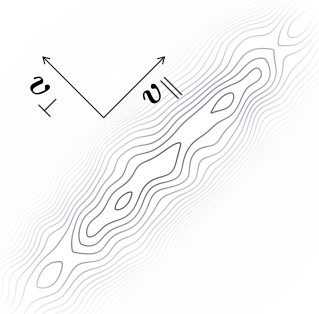

*Figure 6.* Visualization of our argument for uncovering the SADs. We show contours of a hypothetical landscape, $\log q_{\boldsymbol{\theta}, \sigma}(\boldsymbol{x})$, with $\boldsymbol{v}_{\parallel}$ parallel to an induced manifold and $\boldsymbol{v}_{\perp}$ orthogonal.

To decouple the geometry from the data, one may consider an isotropic probe, e.g., $\mathcal{N}(\mathbf{0}, \sigma_{\mathcal{P}}^2 \boldsymbol{I})$. For simplicity, we default to $\mathcal{P} = \delta_{\mathbf{0}} \times \mathcal{U}(\{\sigma_{\min}, \ldots, \sigma_{\max}\})$, i.e., we fix $\boldsymbol{x} = \mathbf{0}$ and draw $\sigma$ uniformly from the noise levels of interest. We shall write the geometry under this default probe as $\mathbf{G}_{\mathcal{F}}$, where the parameters, $\boldsymbol{\theta}$, of $\mathcal{F}$ are assumed to be drawn from the default initialization scheme for the architecture.

Our expectation is that $\mathbf{G}_{\mathcal{F}}$ captures meaningful invariants tied only to the architecture, such as the quirks of iDDPM U-Nets discussed in Section 3.1, that therefore survive and persist in the optimization dynamics of DSM. In particular, having formalized the concept of output-space geometry, we are now ready to state our main conjecture regarding the SADs, linking them to the average geometry at initialization.

*Conjecture* 1. Let $\mathcal{F}$ be a family of networks with geometry $\mathbf{G}_{\mathcal{F}}$. We hypothesize that the eigenvectors of $\mathbf{G}_{\mathcal{F}}$, in *ascending* eigenvalue order, are the SADs. That is, we expect that data aligned with eigenvectors corresponding to small eigenvalues is better modeled compared to data that is aligned with large-eigenvalue eigenvectors.

In the remainder of the paper, we provide empirical evidence to justify Conjecture 1. First, we verify our claims on the rank-one datasets introduced in the beginning of Section 3 in Figure 5, where we draw directions, $\boldsymbol{v} \in \mathbb{S}^{D-1}$, from the eigenvectors of $\mathbf{G}_{\mathcal{F}}$ and benchmark via Wasserstein metrics.

On the left in Figure 5, we focus on iDDPM U-Nets (Nichol & Dhariwal, 2021), which are representative of convolutional diffusion models. The experiments show a clear trend in support of the conjecture, where eigenvectors corresponding to small eigenvalues achieve the best performance and large-eigenvalue vectors have the worst performance. Interestingly, with reference to the visualizations below the plots, we also observe harmonic patterns in the eigenvectors, where large eigenvalues correspond to low frequencies and small eigenvalues to high frequencies. In this sense, our findings provide further evidence in support of the theory of GAHBs (Kadkhodaie et al., 2024), which posits that harmonic representations are fundamental in convolutional models. In particular, our experiment in Figure 7 largely reproduces previously observed patterns. Crucially, beyond the harmonic structure that only loosely characterizes the inductive biases, our notion of geometry is flexible as it is adaptive to architectural details. This is evident in the first eigenvectors, $\boldsymbol{u}_1$, when we ablate the resampling method, also investigated previously in the experiment of Figure 3.

As further evidence of the flexibility of our framework, we now focus our analysis on the DiT architecture (Peebles & Xie, 2023), which is representative of transformer-based diffusion models. The results, shown on the right in Figure 5, are also in support of Conjecture 1. However, as also noted in An et al. (2025), the transformer architecture does not appear to induce harmonic representations. It is also interesting to observe that DiT geometry exhibits considerable eigen-multiplicity, with larger patch sizes amplifying this. We interpret this as evidence of looser structure and weaker inductive biases compared to convolutional networks.

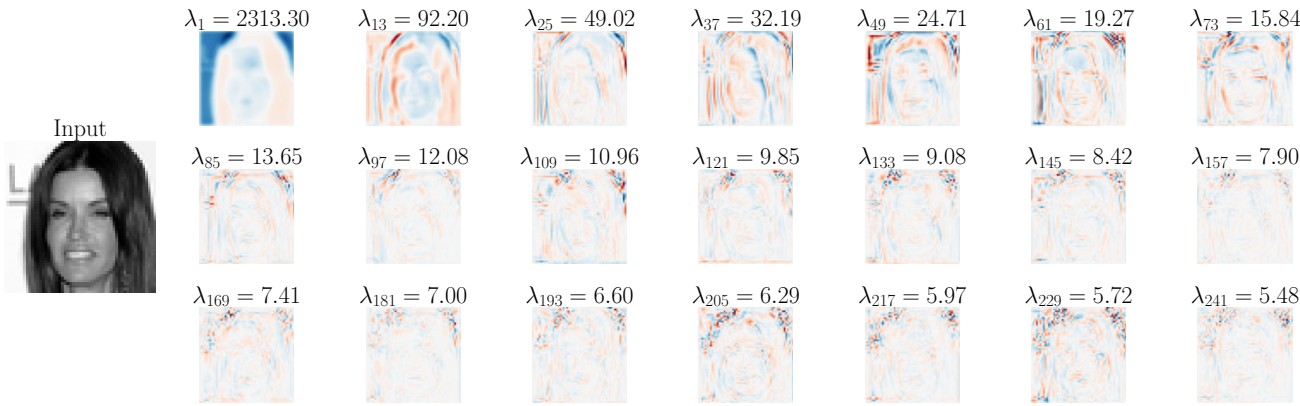

*Figure 7.* Eigendecomposition of iDDPM average geometry, $\mathbf{G}_\mathcal{F}(\mathcal{P}, \Theta)$, at initialization with probing distribution $\mathcal{P} = \mathcal{N}(\boldsymbol{x}, \sigma^2 \boldsymbol{I}) \times \mathcal{U}(\{\sigma_{\min}, \ldots, \sigma_{\max}\})$. That is, we probe along the standard forward diffusion process with the data sample, $\boldsymbol{x}$, shown on the left. The first few eigenvectors of the geometry, together with the corresponding unnormalized eigenvalues, are shown on the right. We note that the results resemble the GAHBs of Kadkhodaie et al. (2024). Specifically, our eigenvectors are also adaptive to the geometry of the input image. Moreover, they have harmonic structure as we observe oscillating patterns whose frequency increases as the eigenvalue decreases.

In fact, Proposition 3 shows that the number of transformer tokens, $T$, is an upper bound on the number of distinct eigenvalues of $\mathbf{G}_\mathcal{F}$. In general, we refer the reader to Appendix B for an analytical treatment of geometries of common architectures and we note that a surprising amount of structure can be inferred simply by inspecting the output layers.

### 3.3. Score Anisotropy Directions in the Wild

Having identified the preferred modeling directions in our experiments on rank-one datasets, we now turn to more realistic data distributions that are encountered in practice. Based on our analysis in Section 3.2, we hypothesize that generalization is largely determined by the (mis)alignment of the data with the average geometry at initialization. For an arbitrary data distribution, $p$, we can extend our setup by defining the alignment with the network, $\alpha$, as follows:

$$\alpha := \mathbb{E}_{\boldsymbol{x} \sim p}[\boldsymbol{z}^\top \mathbf{G}_\mathcal{F} \boldsymbol{z}], \quad \boldsymbol{z} = \boldsymbol{W} \boldsymbol{x}, \quad \boldsymbol{W}^\top \boldsymbol{W} = \boldsymbol{I}. \quad (6)$$

Note, in order to vary $\alpha$ in a way that preserves underlying structure, we introduce the orthogonal matrix $\boldsymbol{W} \in \mathbb{R}^{D \times D}$, which models simple and lossless data transformations, effectively defining a linear autoencoder and a kind of "latent" diffusion model that operates on the transformed data. With this setup, our Conjecture 1 amounts to the claim that the best performance is observed when $\alpha$ is minimized and the worst when $\alpha$ is maximized. The corresponding orthogonal matrices, $\boldsymbol{W}_{\min}$ and $\boldsymbol{W}_{\max}$, that achieve such extreme (mis)alignment are given by Theorem 2.

**Theorem 2** (Extreme alignment, proof in Appendix C.5).
*Let $\boldsymbol{U}$, $\boldsymbol{V}$ be matrices whose columns are eigenvectors of $\mathbf{G}_\mathcal{F}$, $\mathbb{E}_{\boldsymbol{x} \sim p}[\boldsymbol{x} \boldsymbol{x}^\top]$ respectively, ordered according to their associated eigenvalues from largest to smallest. Let $\boldsymbol{J}$ denote the row-reversed identity matrix. Then, $\alpha$ is minimized for $\boldsymbol{W}_{\min} = \boldsymbol{U} \boldsymbol{J} \boldsymbol{V}^\top$ and maximized for $\boldsymbol{W}_{\max} = \boldsymbol{U} \boldsymbol{V}^\top$.*

To test our hypothesis, we train such latent diffusion models under identical settings for each $\boldsymbol{W}$. We also consider a baseline corresponding to $\boldsymbol{W} = \boldsymbol{I}$, i.e., the natural alignment of the data with the average geometry. In particular, we conduct experiments on the MNIST (LeCun et al., 1998) ($28 \times 28$), CelebA-HQ (Karras et al., 2018) ($56 \times 56$) and CIFAR-10 (Krizhevsky, 2009) ($32 \times 32$) datasets, with implementation details included in Appendix A.

Our findings, shown in Figures 8 and 9, broadly agree with our central conjecture. For example, focusing on the MNIST samples obtained by our models, shown in Figure 8, we observe that large $\alpha$ results in artifacts at the distribution level.[2] Specifically, for the default alignment, which is already significant, a considerable fraction of samples do not contain a digit. Moreover, explicitly maximizing $\alpha$ appears to lead to mode collapse, with the digit "1" being overrepresented. When we instead minimize $\alpha$, substantial and consistent improvements are observed across all datasets, as also quantified by Wasserstein metrics in Figure 9.

## 4. Discussion

We presented and empirically validated a framework for determining architectural directional biases of diffusion models. Quite surprisingly, we find that, despite the highly non-linear nature of modern neural networks, these biases are well-described via fixed bases that may be decoupled from the data, making them useful for explaining and predicting generalization ability, as quantified via Wasserstein metrics. We conclude with a discussion on potential future applications of our proposed framework, related work on inductive biases and limitations of our study.

---

[2] In general, we do not expect perceptual quality to correlate with Wasserstein distances.

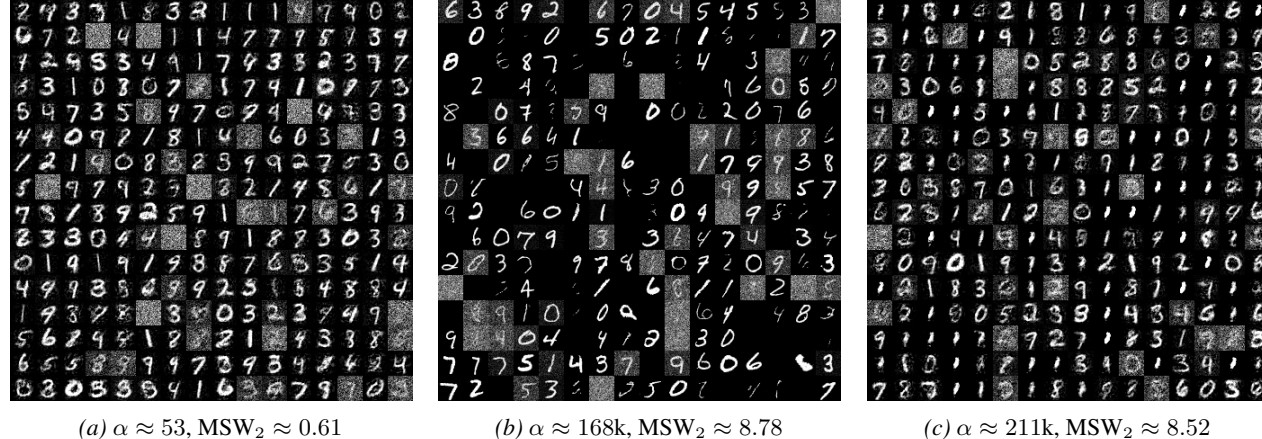

*(a)* $\alpha \approx 53$, $\mathrm{MSW}_2 \approx 0.61$     *(b)* $\alpha \approx 168\mathrm{k}$, $\mathrm{MSW}_2 \approx 8.78$     *(c)* $\alpha \approx 211\mathrm{k}$, $\mathrm{MSW}_2 \approx 8.52$

*Figure 8.* Uncurated samples of MNIST-trained iDDPMs. We vary the alignment of the data with the geometry, $\alpha := \mathbb{E}_{\boldsymbol{x} \sim p}[\boldsymbol{z}^\top \mathbf{G}_{\mathcal{F}} \boldsymbol{z}]$, by transforming $p$ via appropriate orthogonal matrices $\boldsymbol{W}$. On the left, we show the effect of minimizing $\alpha$, where the model gives a reasonable approximation of the ground truth distribution. The middle shows the default alignment, i.e., we do not apply any transformation. As also quantified via the Wasserstein distances, it is evident that the data distribution is not well-modeled in this case as a considerable fraction of samples do not contain a digit. The right shows samples from the model corresponding to maximizing $\alpha$, where we see similar $\mathrm{MSW}_2$ to the middle. Interestingly, this last experiment suggests mode collapse, where the model is more likely to generate the digit "1".

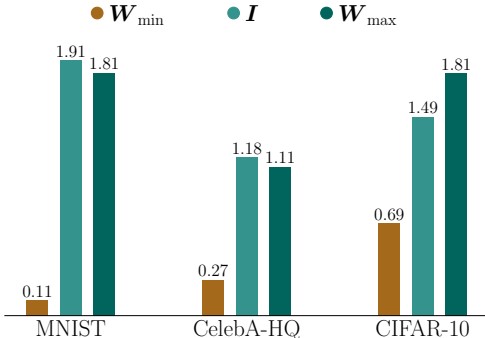

*Figure 9.* iDDPM average performance ($\mathrm{SW}_2$) on standard image datasets as the alignment with the geometry, $\alpha$, varies. We show the effect of minimizing, maximizing $\alpha$ as well as the default setting. These correspond to matrices $\boldsymbol{W}_{\min}$, $\boldsymbol{W}_{\max}$ and $\boldsymbol{I}$ respectively.

Regarding future applications, much of modern deep learning is empirical, enabled by scale and guided by heuristics. Our insights and further developments along this line of research could allow for more cost-effective and principled development of generative technologies. For example, we see potential applications in AutoML (Bergstra et al., 2011) and neural architecture search (Zoph & Le, 2017).

Importantly, characterizing inductive biases of generative models has implications for understanding and mitigating undesirable behaviors such as memorization of training data (Carlini et al., 2023; Somepalli et al., 2023; Gu et al., 2025) and hallucination in generations (Aithal et al., 2024; Lu et al., 2025; Floros et al., 2025). By making explicit which patterns a model is predisposed to reproduce, our approach could help identify circumstances under which models are likely to overfit or generate spurious content.

## 4.1. Related Work

Kadkhodaie et al. (2024) observe that convolutional diffusion is biased toward GAHBs. They give a shrinkage-based interpretation of denoising with adaptive eigenbases defined by local Jacobians of trained networks. A similar Jacobian-based analysis was explored in transformers by An et al. (2025). However, no obvious regularities in the eigenbases were found that could be systematically exploited. Instead, SADs provide a more unified characterization of biases.

With some variations, a notion similar to our average geometry previously appeared in Ortiz-Jimenez et al. (2020); Movahedi et al. (2025), who study directional inductive biases of discriminative networks. Interestingly, their analysis predicts classifiers actually perform better when data is aligned with the *largest* eigenvalues of the geometry. Although their setup is not identical to ours, and therefore not directly comparable, these seemingly contrasting conclusions motivate further investigation, left for future work.

## 4.2. Limitations

Our experiments focus on relatively small-scale settings. A rigorous validation of our claims at large scales would require significantly more resources and time. Ultimately, we chose to prioritize insight as opposed to completeness. Moreover, it is important to stress that we have specifically isolated directional inductive biases imposed by the architecture. In principle, the overall dynamics of diffusion models may be influenced or dominated by different factors such as explicit regularization or other implicit priors. In general, we do not make claims regarding foundation models nor do we claim a complete theory of inductive biases.

## Impact Statement

This paper presents work whose goal is to advance the field of Machine Learning. There are many potential societal consequences of our work, none which we feel must be specifically highlighted here.

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

*Table 1.* Hyperparameters of iDDPM and DiT architectures used in the paper. *For the same number of iterations, DiT/4 networks converge slower compared to DiT/2 but have shorter training times. We doubled their iterations since this was relatively inexpensive.

| iDDPM | $\mathcal{N}(\mathbf{0}, D\boldsymbol{vv}^\top)$ | MNIST | CelebA-HQ | CIFAR-10 | DiT | $\mathcal{N}(\mathbf{0}, D\boldsymbol{vv}^\top)$ | Sphere |
|---|---|---|---|---|---|---|---|
| shape | $1 \times 16 \times 16$ | $1 \times 28 \times 28$ | $1 \times 56 \times 56$ | $3 \times 32 \times 32$ | shape | $1 \times 16 \times 16$ | $1 \times 16 \times 16$ |
| diffusion_steps | 1000 | 1000 | 1000 | 1000 | diffusion_steps | 1000 | 1000 |
| noise_schedule | linear | linear | linear | linear | noise_schedule | linear | linear |
| channels | 32 | 32 | 32 | 32 | hidden_size | 48 | 48 |
| channel_mults | 1, 1 | 1, 1 | 1, 1, 1 | 1, 1, 1 | patch_size | 2/4 | 2 |
| depth | 1 | 2 | 2 | 2 | depth | 8 | 8 |
| attn_resolutions | - | - | - | - | mlp_ratio | 2 | 2 |
| num_heads | 4 | 4 | 4 | 4 | num_heads | 4 | 4 |
| batch_size | 1000 | 500 | 500 | 500 | batch_size | 1000 | 1000 |
| iterations | 2k | 100k | 100k | 200k | iterations | 2/4k* | 10k |
| learning_rate | 1e-4 | 1e-3 | 1e-3 | 1e-3 | learning_rate | 1e-4 | 1e-4 |

# A. Experimental Setup

All of our experiments were conducted on a Linux machine with a NVIDIA RTX 4090 GPU. We train and evaluate diffusion models according to the DDPM framework (Ho et al., 2020), with our hyperparameters included in Table 1. To compute the (M)SW$_2$ metrics we use an overcomplete set of $L = 64D$ random directions. Each dataset we consider consists of 10k training and 10k testing samples. For sphere modeling, shown in Figure 1, data is aligned with the last three and the first three SADs. When estimating the average geometry, $\mathbf{G}_\mathcal{F}(\mathcal{P}, \Theta)$, we use 1M randomly initialized networks.

# B. Geometries of Common Neural Network Architectures

**Proposition 1** (MLP geometry, proof in Appendix C.1)**.** *Let $\mathcal{F}$ be networks of the form $\boldsymbol{z} = \phi(\boldsymbol{Wh} + \boldsymbol{b})$, where $\phi : \mathbb{R} \to \mathbb{R}$ is element-wise and $\boldsymbol{h} \in \mathbb{R}^L$ is some function of the input. Assume that $\boldsymbol{W}, \boldsymbol{b}$ are completely independent and parameters within each group are identically distributed. Letting $\mathbf{1}$ represent a vector of ones, the geometry takes the form:*

$$\mathbf{G}_\mathcal{F}(\mathcal{P}, \Theta) = \alpha_\mathcal{F}(\mathcal{P}, \Theta)\boldsymbol{I} + \beta_\mathcal{F}(\mathcal{P}, \Theta)\mathbf{1}\mathbf{1}^\top. \tag{7}$$

**Proposition 2** (CNN geometry, proof in Appendix C.2)**.** *Let $\mathcal{F}$ be the family of networks of the form $\boldsymbol{z} = \phi(\boldsymbol{Wh} + \boldsymbol{b})$, where $\phi : \mathbb{R} \to \mathbb{R}$ is element-wise and $\boldsymbol{h} \in \mathbb{R}^{C_{in} \times L}$ denotes some function of the input. $\boldsymbol{W}$ represents convolution with kernel size $k$, $C_{in}$ input channels and $C_{out}$ output channels. Assume $\boldsymbol{W}, \boldsymbol{b}$ are completely independent and parameters within each group are identically distributed. Letting $\otimes$ denote the Kronecker product, the geometry takes the form:*

$$\mathbf{G}_\mathcal{F}(\mathcal{P}, \Theta) = \boldsymbol{I}_{C_{out}} \otimes \boldsymbol{A}_\mathcal{F}(\mathcal{P}, \Theta) + (\mathbf{1}\mathbf{1}^\top)_{C_{out}} \otimes \boldsymbol{B}_\mathcal{F}(\mathcal{P}, \Theta). \tag{8}$$

**Proposition 3** (Transformer geometry, proof in Appendix C.3)**.** *Let $\mathcal{F}$ be networks of the form $\boldsymbol{z} = \boldsymbol{Q}(\boldsymbol{Wh} + \boldsymbol{b})$, where $\boldsymbol{Q}$ is fixed, orthonormal (e.g., unpatchify) and $\boldsymbol{h} \in \mathbb{R}^{T \times L_{in}}$ is a function of the input. $\boldsymbol{W} : \mathbb{R}^{T \times L_{in}} \to \mathbb{R}^{T \times L_{out}}$ and $\boldsymbol{b}$ represent a transformer layer operating on $T$ tokens separately. Assume $\boldsymbol{W}, \boldsymbol{b}$ are completely independent and parameters within each group are zero-mean, identically distributed. The geometry has at most $T$ distinct eigenvalues and takes the form:*

$$\mathbf{G}_\mathcal{F}(\mathcal{P}, \Theta) = \boldsymbol{Q}[\boldsymbol{A}_\mathcal{F}(\mathcal{P}, \Theta) \otimes \boldsymbol{I}_{L_{out}}]\boldsymbol{Q}^\top. \tag{9}$$

## C. Deferred Proofs

**Lemma 1.** *Consider the DSM setup with data drawn from $\mathcal{N}(\mathbf{0}, \boldsymbol{v}\boldsymbol{v}^\top)$ for a fixed noise level $\sigma > 0$ and with $\boldsymbol{v} \in \mathbb{S}^{D-1}$. The associated optimal score function is linear and of the form $\boldsymbol{\Omega}(\cdot)$, with $\boldsymbol{\Omega} = \frac{1}{\sigma^2}\left(\frac{\boldsymbol{v}\boldsymbol{v}^\top}{\sigma^2+1} - \boldsymbol{I}\right).$*

*Proof.* $q_\sigma = \mathcal{N}(\mathbf{0}, \boldsymbol{v}\boldsymbol{v}^\top + \sigma^2\boldsymbol{I})$. The log-density is therefore quadratic, resulting in a linear score function as follows:

$$\nabla_{\boldsymbol{x}} \log q_\sigma(\boldsymbol{x}) = -\frac{1}{2}\nabla_{\boldsymbol{x}}\left[\boldsymbol{x}^\top(\boldsymbol{v}\boldsymbol{v}^\top + \sigma^2\boldsymbol{I})^{-1}\boldsymbol{x}\right] = -(\boldsymbol{v}\boldsymbol{v}^\top + \sigma^2\boldsymbol{I})^{-1}\boldsymbol{x}. \tag{10}$$

We can further simplify this via the Sherman-Morrison formula:

$$\boldsymbol{\Omega} = -(\boldsymbol{v}\boldsymbol{v}^\top + \sigma^2\boldsymbol{I})^{-1} = \frac{1}{\sigma^2}\left(\frac{\boldsymbol{v}\boldsymbol{v}^\top}{\sigma^2+1} - \boldsymbol{I}\right). \tag{11}$$

$\square$

**Lemma 2.** *Let $\boldsymbol{W}$ be a random matrix with entries that are iid. Assume that their mean is zero and their variance is $\sigma^2$. For any compatible matrix, $\boldsymbol{X}$, independent of $\boldsymbol{W}$, we have $\mathbb{E}_{\boldsymbol{W}}[\boldsymbol{W}^\top\boldsymbol{X}\boldsymbol{W}] = \sigma^2\operatorname{tr}(\boldsymbol{X})\boldsymbol{I}$, where $\operatorname{tr}(\cdot)$ denotes the trace.*

*Proof.* Let $(\cdot)^{(i)}$ be column $i$. For entry $(i, j)$ we can write the expectation as follows:

$$\sum_k \mathbb{E}\left[\boldsymbol{W}^{(i)^\top}\boldsymbol{X}^{(k)}\boldsymbol{W}_{(k,j)}\right] = \sum_k \sigma^2\delta_{i-j}\boldsymbol{X}_{(k,k)} = \sigma^2\operatorname{tr}(\boldsymbol{X})\delta_{i-j}. \tag{12}$$

$\square$

### C.1. Proposition 1

*Proof.* Let $(\cdot)^{(i)}$ be row $i$. The average geometry is expressed as:

$$\mathbf{G}_{\mathcal{F}}(\mathcal{P}, \Theta) = \mathbb{E}_{\boldsymbol{h}} \mathbb{E}_{\boldsymbol{W}, \boldsymbol{b}} \left[ \phi(\boldsymbol{W}\boldsymbol{h} + \boldsymbol{b}) \phi(\boldsymbol{W}\boldsymbol{h} + \boldsymbol{b})^{\top} \right]. \tag{13}$$

First, compute the inner expectation, i.e., $\mathbf{G}_{\mathcal{F}}(\mathcal{P}, \Theta)|_{\boldsymbol{h}}$. For indices $i, j$, since the parameters are iid:

$$[\mathbf{G}_{\mathcal{F}}(\mathcal{P}, \Theta)|_{\boldsymbol{h}}]_{(i,j)} = \begin{cases} \mathbb{E}_{\boldsymbol{W}, \boldsymbol{b}}[\phi(\boldsymbol{W}^{(1)}\boldsymbol{h} + \boldsymbol{b}^{(1)})^2] & \text{if } i = j \\ \mathbb{E}_{\boldsymbol{W}, \boldsymbol{b}}[\phi(\boldsymbol{W}^{(1)}\boldsymbol{h} + \boldsymbol{b}^{(1)})]^2 & \text{otherwise} \end{cases}. \tag{14}$$

Writing $\boldsymbol{z}^{(1)} = \phi(\boldsymbol{W}^{(1)}\boldsymbol{h} + \boldsymbol{b}^{(1)})$, we express the above via the conditional mean, $\mu_{\boldsymbol{z}^{(1)}|\boldsymbol{h}}$, and conditional variance $\sigma^2_{\boldsymbol{z}^{(1)}|\boldsymbol{h}}$:

$$\mathbf{G}_{\mathcal{F}}(\mathcal{P}, \Theta)|_{\boldsymbol{h}} = \sigma^2_{\boldsymbol{z}^{(1)}|\boldsymbol{h}} \boldsymbol{I} + \mu^2_{\boldsymbol{z}^{(1)}|\boldsymbol{h}} \mathbf{1}\mathbf{1}^{\top}, \tag{15}$$

where $\mathbf{1}$ is a vector of ones. Now, taking the outer expectation yields the desired result:

$$\mathbf{G}_{\mathcal{F}}(\mathcal{P}, \Theta) = \mathbb{E}_{\boldsymbol{h}}[\sigma^2_{\boldsymbol{z}^{(1)}|\boldsymbol{h}}]\boldsymbol{I} + \mathbb{E}_{\boldsymbol{h}}[\mu^2_{\boldsymbol{z}^{(1)}|\boldsymbol{h}}]\mathbf{1}\mathbf{1}^{\top}. \tag{16}$$

$\square$

### C.2. Proposition 2

*Proof.* As in our treatment of MLPs in Appendix C.1, we first compute the geometry conditioned on $\boldsymbol{h}$, i.e., $\mathbf{G}(\mathcal{P}, \Theta)|_{\boldsymbol{h}}$. Vectorizing, write the $i^{\text{th}}$ input channel as $\boldsymbol{h}^{(i)} \in \mathbb{R}^L$. Then, $\boldsymbol{W}$ is a $C_{\text{out}} \times C_{\text{in}}$ block matrix with each $\boldsymbol{W}_{(i,j)}$ representing convolution with a filter of size $k$. Therefore, we write $\boldsymbol{z}^{(i)} = \phi(\sum_j \boldsymbol{W}_{(i,j)}\boldsymbol{h}^{(j)} + \boldsymbol{b}^{(i)})$. Since the parameters are iid, the conditional geometry is expressed as the following $C_{\text{out}} \times C_{\text{out}}$ block matrix:

$$[\mathbf{G}_{\mathcal{F}}(\mathcal{P}, \Theta)|_{\boldsymbol{h}}]_{(m,n)} = \begin{cases} \mathbb{E}_{\boldsymbol{W}, \boldsymbol{b}}[\boldsymbol{z}^{(1)}\boldsymbol{z}^{(1)\top}] & \text{if } m = n \\ \mathbb{E}_{\boldsymbol{W}, \boldsymbol{b}}[\boldsymbol{z}^{(1)}]\mathbb{E}_{\boldsymbol{W}, \boldsymbol{b}}[\boldsymbol{z}^{(1)}]^{\top} & \text{otherwise} \end{cases}. \tag{17}$$

We can rephrase the above result in terms of the conditional mean, $\boldsymbol{\mu}_{\boldsymbol{z}^{(1)}|\boldsymbol{h}}$, and a conditional covariance matrix $\boldsymbol{\Sigma}_{\boldsymbol{z}^{(1)}|\boldsymbol{h}}$. The manipulation is identical to the one used to derive the MLP geometry. Averaging over $\boldsymbol{h}$ yields:

$$\mathbf{G}_{\mathcal{F}}(\mathcal{P}, \Theta) = \boldsymbol{I}_{C_{\text{out}}} \otimes \mathbb{E}_{\boldsymbol{h}}[\boldsymbol{\Sigma}_{\boldsymbol{z}^{(1)}|\boldsymbol{h}}] + (\mathbf{1}\mathbf{1}^{\top})_{C_{\text{out}}} \otimes \mathbb{E}_{\boldsymbol{h}}[\boldsymbol{\mu}_{\boldsymbol{z}^{(1)}|\boldsymbol{h}}\boldsymbol{\mu}^{\top}_{\boldsymbol{z}^{(1)}|\boldsymbol{h}}]. \tag{18}$$

$\square$

### C.3. Proposition 3

*Proof.* We first focus on $\boldsymbol{Q} = \boldsymbol{I}$. After vectorizing $\boldsymbol{h}$, $\boldsymbol{W}$ is block-diagonal with $\boldsymbol{W}_{(i,i)} = \boldsymbol{W}_{(1,1)}$ operating on each token independently, which we write as $\boldsymbol{h}^{(i)} \in \mathbb{R}^{L_{\text{in}}}$. Conditioned on $\boldsymbol{h}$, the geometry is a block matrix with block $(i,j)$:

$$\begin{aligned} [\mathbf{G}_{\mathcal{F}}(\mathcal{P}, \Theta)|_{\boldsymbol{h}}]_{(i,j)} &= \mathbb{E}_{\boldsymbol{W}, \boldsymbol{b}}[(\boldsymbol{W}_{(1,1)}\boldsymbol{h}^{(i)} + \boldsymbol{b}^{(1)})(\boldsymbol{W}_{(1,1)}\boldsymbol{h}^{(j)} + \boldsymbol{b}^{(1)})^{\top}] \\ &= (\sigma^2_{\boldsymbol{W}}\boldsymbol{h}^{(i)\top}\boldsymbol{h}^{(j)} + \sigma^2_{\boldsymbol{b}})\boldsymbol{I}, \end{aligned} \tag{19}$$

where the last equality is by Lemma 2, assuming parameters in $\boldsymbol{W}, \boldsymbol{b}$ have variances $\sigma^2_{\boldsymbol{W}}, \sigma^2_{\boldsymbol{b}}$ respectively. Note, every block is $\propto \boldsymbol{I}$ and depends on entries of $\boldsymbol{h}\boldsymbol{h}^{\top} \in \mathbb{R}^{T \times T}$, where, with a slight abuse of notation, we have reverted to the matrix representation $\boldsymbol{h} \in \mathbb{R}^{T \times L_{in}}$. Writing this compactly as $(\sigma^2_{\boldsymbol{W}}\boldsymbol{h}\boldsymbol{h}^{\top} + \sigma^2_{\boldsymbol{b}}\mathbf{1}\mathbf{1}^{\top}) \otimes \boldsymbol{I}_{L_{\text{out}}}$, applying $\boldsymbol{Q}$ and averaging over $\boldsymbol{h}$:

$$\mathbf{G}_{\mathcal{F}}(\mathcal{P}, \Theta) = \boldsymbol{Q}\mathbb{E}_{\boldsymbol{h}}[(\sigma^2_{\boldsymbol{W}}\boldsymbol{h}\boldsymbol{h}^{\top} + \sigma^2_{\boldsymbol{b}}\mathbf{1}\mathbf{1}^{\top}) \otimes \boldsymbol{I}_{L_{\text{out}}}]\boldsymbol{Q}^{\top}. \tag{20}$$

Eigenvalues are products of eigenvalues of factors in the Kronecker product, i.e., the $T$ eigenvalues of $\sigma^2_{\boldsymbol{W}}\mathbb{E}_{\boldsymbol{h}}[\boldsymbol{h}\boldsymbol{h}^{\top}] + \sigma^2_{\boldsymbol{b}}\mathbf{1}\mathbf{1}^{\top}$.

$\square$

## C.4. Theorem 1

*Proof.* Let $\boldsymbol{\Omega} = \boldsymbol{\Phi\Theta}$ be a linear model. The optimization objective is:

$$
\begin{aligned}
\mathcal{J}_{\text{DSM}}(\boldsymbol{\Theta}) &= \mathbb{E}_{\boldsymbol{x}\sim\mathcal{N}(\boldsymbol{0},\boldsymbol{vv}^\top),\boldsymbol{\epsilon}\sim\mathcal{N}(\boldsymbol{0},\boldsymbol{I})}\left[\left\|\boldsymbol{\Phi\Theta}(\boldsymbol{x}+\sigma\boldsymbol{\epsilon})+\frac{\boldsymbol{\epsilon}}{\sigma}\right\|_2^2\right] \\
&= \mathbb{E}\left[\left\|\boldsymbol{\Phi\Theta}\boldsymbol{x}+\left(\sigma\boldsymbol{\Phi\Theta}+\frac{\boldsymbol{I}}{\sigma}\right)\boldsymbol{\epsilon}\right\|_2^2\right] \\
&\overset{(*)}{=} \mathbb{E}\left[\|\boldsymbol{\Phi\Theta}\boldsymbol{x}\|_2^2\right]+\mathbb{E}\left[\left\|\left(\sigma\boldsymbol{\Phi\Theta}+\frac{\boldsymbol{I}}{\sigma}\right)\boldsymbol{\epsilon}\right\|_2^2\right] = \|\boldsymbol{\Phi\Theta v}\|_2^2+\left\|\sigma\boldsymbol{\Phi\Theta}+\frac{\boldsymbol{I}}{\sigma}\right\|_F^2,
\end{aligned}
\tag{21}
$$

where $(*)$ follows by independence of $\boldsymbol{x}$ and $\boldsymbol{\epsilon}$. The gradient with respect to $\boldsymbol{\Theta}$ is then given by:

$$
\nabla_{\boldsymbol{\Theta}}\mathcal{J}_{\text{DSM}}(\boldsymbol{\Theta}) = 2\boldsymbol{\Phi}^\top[\boldsymbol{\Phi\Theta}(\boldsymbol{vv}^\top+\sigma^2\boldsymbol{I})+\boldsymbol{I}].
\tag{22}
$$

With this, and for a suitable $\eta > 0$, we express the GD learning dynamics with respect to $\boldsymbol{\Omega}$ as:

$$
\boldsymbol{\Omega}_t = \boldsymbol{\Omega}_{t-1} - \eta\boldsymbol{\Phi}\nabla_{\boldsymbol{\Theta}}\mathcal{J}_{\text{DSM}}(\boldsymbol{\Theta}) = \boldsymbol{\Omega}_{t-1} - 2\eta\boldsymbol{\Phi\Phi}^\top[\boldsymbol{\Omega}_{t-1}(\boldsymbol{vv}^\top+\sigma^2\boldsymbol{I})+\boldsymbol{I}].
\tag{23}
$$

Moreover, we study the error dynamics defined by $\boldsymbol{E}_t = \boldsymbol{\Omega}_t - \boldsymbol{\Omega}^*$. Here, $\boldsymbol{\Omega}^* = \frac{1}{\sigma^2}\left(\frac{\boldsymbol{vv}^\top}{\sigma^2+1}-\boldsymbol{I}\right)$ is given by Lemma 1 and we can verify that $\boldsymbol{\Omega}^*(\boldsymbol{vv}^\top+\sigma^2\boldsymbol{I})+\boldsymbol{I} = \boldsymbol{0}$, so it is a stationary point. Therefore, we write:

$$
\boldsymbol{E}_t = \boldsymbol{E}_{t-1} - 2\eta\boldsymbol{\Phi\Phi}^\top\boldsymbol{E}_{t-1}(\boldsymbol{vv}^\top+\sigma^2\boldsymbol{I}).
\tag{24}
$$

Focus on the sequence of expected errors, $\mathbb{E}[\boldsymbol{E}_t]$, where the randomness is over the initialization.[3] Assuming $\mathbb{E}[\boldsymbol{\Theta}_0] = \boldsymbol{0}$, $\mathbb{E}[\boldsymbol{E}_0] = -\boldsymbol{\Omega}^*$. Additionally, if $\boldsymbol{v} \in \{\boldsymbol{u}_i\}_{i=1}^D$ is an eigenvector of $\boldsymbol{\Phi\Phi}^\top$ with corresponding eigenvalues $\{\lambda_i\}_{i=1}^D$, matrices in Equation 24 commute since they share eigenvectors. This forces the eigenspaces of all subsequent error terms. We have:

$$
\mathbb{E}[\boldsymbol{E}_t] = \mathbb{E}[\boldsymbol{E}_{t-1}] - 2\eta\boldsymbol{\Phi\Phi}^\top(\boldsymbol{u}_i\boldsymbol{u}_i^\top+\sigma^2\boldsymbol{I})\mathbb{E}[\boldsymbol{E}_{t-1}] = -[\boldsymbol{I} - 2\eta(\lambda_i\boldsymbol{u}_i\boldsymbol{u}_i^\top+\sigma^2\boldsymbol{\Phi\Phi}^\top)]^t\boldsymbol{\Omega}^*,
\tag{25}
$$

where it is clear that the error decays exponentially. In particular, for sufficiently small $\eta$, the iterated matrix is positive semidefinite and therefore convergence depends on the minimum eigenvalue of $\lambda_i\boldsymbol{u}_i\boldsymbol{u}_i^\top+\sigma^2\boldsymbol{\Phi\Phi}^\top$, i.e., it is $\mathcal{O}[(1-2\eta\rho_i)^t]$ with $\rho_i = \min[(\sigma^2+1)\lambda_i, \sigma^2\min_{j\neq i}\lambda_j]$. Suppose $\exists\lambda_j < \lambda_i$, then $\rho_i = \sigma^2\lambda_D$, i.e., the convergence rate is fixed for $i < D$. However, for $i = D$ we have $\rho_D = \min[(\sigma^2+1)\lambda_D, \sigma^2\lambda_{D-1}] > \sigma^2\lambda_D$. That is, we converge faster for $i = D$.

Now, to complete the proof, we focus on the stochastic gradient at optimality:

$$
\nabla_{\boldsymbol{\Theta}}\widehat{\mathcal{J}}_{\text{DSM}}(\boldsymbol{x},\boldsymbol{\epsilon};\boldsymbol{\Theta}^*) = 2\boldsymbol{pq}^\top, \quad \boldsymbol{p} = \boldsymbol{\Phi}^\top\left(\boldsymbol{\Omega}^*\boldsymbol{q}+\frac{\boldsymbol{\epsilon}}{\sigma}\right), \quad \boldsymbol{q} = \boldsymbol{x}+\sigma\boldsymbol{\epsilon}.
\tag{26}
$$

Note, by construction, all of the above quantities are zero-mean. In particular, this implies that $\boldsymbol{p}, \boldsymbol{q}$ are uncorrelated and, since they are jointly Gaussian, we claim that they are independent. Therefore, by setting $\boldsymbol{v} = \boldsymbol{u}_i$ and vectorizing, we can write the stochastic gradient covariance as:

$$
\begin{aligned}
4\mathbb{E}\left[\text{vec}(\boldsymbol{pq}^\top)\text{vec}(\boldsymbol{pq}^\top)^\top\right] &= 4\mathbb{E}\left[(\boldsymbol{qq}^\top)\otimes(\boldsymbol{pp}^\top)\right] \\
&= 4\mathbb{E}\left[\boldsymbol{qq}^\top\right]\otimes\mathbb{E}\left[\boldsymbol{pp}^\top\right] \\
&= \frac{4}{\sigma^2(\sigma^2+1)}(\boldsymbol{u}_i\boldsymbol{u}_i^\top+\sigma^2\boldsymbol{I})\otimes(\boldsymbol{\Phi}^\top\boldsymbol{u}_i\boldsymbol{u}_i^\top\boldsymbol{\Phi}) \propto \lambda_i.
\end{aligned}
\tag{27}
$$

$\square$

---

[3] More generally, our analysis here is also applicable to fluctuations arising due to SGD.

## C.5. Theorem 2

*Proof.* We first simplify Equation 6:

$$\alpha = \mathbb{E}_{\boldsymbol{x} \sim p}[\boldsymbol{x}^\top (\boldsymbol{W}^\top \mathbf{G}_{\mathcal{F}} \boldsymbol{W} \boldsymbol{x})] = \mathrm{tr}(\boldsymbol{W}^\top \mathbf{G}_{\mathcal{F}} \boldsymbol{W} \boldsymbol{C}), \tag{28}$$

where $\boldsymbol{C} := \mathbb{E}_{\boldsymbol{x} \sim p}[\boldsymbol{x}\boldsymbol{x}^\top]$ is the second moment of the data. Let $\mathbf{G}_{\mathcal{F}} = \boldsymbol{U}\boldsymbol{\Lambda}\boldsymbol{U}^\top$ and $\boldsymbol{C} = \boldsymbol{V}\boldsymbol{\Sigma}\boldsymbol{V}^\top$ be eigendecompositions with eigenvalues $\{\lambda_i\}_{i=1}^D$ and $\{\sigma_i\}_{i=1}^D$ respectively. Define the matrix $\boldsymbol{Q} = \boldsymbol{U}^\top \boldsymbol{W} \boldsymbol{V}$. Then, Equation 28 is equivalent to:

$$\begin{aligned}
\alpha &= \mathrm{tr}(\boldsymbol{W}^\top \boldsymbol{U}\boldsymbol{\Lambda}\boldsymbol{U}^\top \boldsymbol{W}\boldsymbol{V}\boldsymbol{\Sigma}\boldsymbol{V}^\top) \\
&= \mathrm{tr}(\boldsymbol{V}^\top \boldsymbol{W}^\top \boldsymbol{U}\boldsymbol{\Lambda}\boldsymbol{U}^\top \boldsymbol{W}\boldsymbol{V}\boldsymbol{\Sigma}) \\
&= \mathrm{tr}(\boldsymbol{Q}^\top \boldsymbol{\Lambda}\boldsymbol{Q}\boldsymbol{\Sigma}) = \sum_{i,j} \lambda_i \sigma_j [\boldsymbol{Q}_{(i,j)}]^2.
\end{aligned} \tag{29}$$

Since $\boldsymbol{Q}$ is orthogonal, the matrix defined by $\boldsymbol{P}_{(i,j)} = [\boldsymbol{Q}_{(i,j)}]^2$ is doubly stochastic. Moreover, Equation 29 shows optimizing $\alpha$ is linear in $\boldsymbol{P}$ over the set of doubly stochastic matrices. In particular, by Birkhoff's theorem, $\alpha$ is extremized when $\boldsymbol{P}$ is a permutation. The minimizer is the one that misaligns eigenvalues, i.e., $\boldsymbol{P} = \boldsymbol{J}$ is achieved if $\boldsymbol{Q} = \boldsymbol{J} \iff \boldsymbol{W} = \boldsymbol{U}\boldsymbol{J}\boldsymbol{V}^\top$. Similarly, $\alpha$ is maximized when eigenvalues are aligned, i.e., $\boldsymbol{P} = \boldsymbol{I}$ if $\boldsymbol{Q} = \boldsymbol{I} \iff \boldsymbol{W} = \boldsymbol{U}\boldsymbol{V}^\top$. Indeed, these permutations are optimal as they satisfy von Neumann's trace inequalities with equality (cf. Ruhe). $\square$

